# A 2000-Year-Old *Bacillus stercoris* Strain Sheds Light on the Evolution of Cyclic Antimicrobial Lipopeptide Synthesis

**DOI:** 10.3390/microorganisms12020338

**Published:** 2024-02-06

**Authors:** Bessem Chouaia, Jessica Dittmer

**Affiliations:** 1Department of Environmental Sciences, Informatics and Statistics, Ca’ Foscari University of Venice, 30172 Venice, Italy; 2Faculty of Agricultural, Environmental and Food Sciences, Free University of Bozen-Bolzano, 39100 Bolzano, Italy; jessica.dittmer@inrae.fr; 3UMR 1345, Institut Agro, INRAE, IRHS, SFR Quasav, Université d’Angers, 49070 Beaucouzé, France

**Keywords:** *Bacillus stercoris*, ancient bacteria, surfactin

## Abstract

Some bacteria (notably the genera *Bacillus* and *Clostridium*) have the capacity to form endospores that can survive for millions of years in isolated habitats. The genomes of such ancient bacteria provide unique opportunities to understand bacterial evolution and metabolic capabilities over longer time scales. Herein, we sequenced the genome of a 2000-year-old bacterial strain (Mal05) isolated from intact apple seeds recovered during archaeological excavations of a Roman villa in Italy. Phylogenomic analyses revealed that this strain belongs to the species *Bacillus stercoris* and that it is placed in an early-branching position compared to most other strains of this species. Similar to other *Bacillus* species, *B. stercoris* Mal05 had been previously shown to possess antifungal activity. Its genome encodes all the genes necessary for the biosynthesis of fengycin and surfactin, two cyclic lipopeptides known to play a role in the competition of Bacilli with other microorganisms due to their antimicrobial activity. Comparative genomics and analyses of selective pressure demonstrate that these genes are present in all sequenced *B. stercoris* strains, despite the fact that they are not under strong purifying selection. Hence, these genes may not be essential for the fitness of these bacteria, but they can still provide a competitive advantage against other microorganisms present in the same environment.

## 1. Introduction

Some bacteria (notably the genera *Bacillus* and *Clostridium*) have the capacity to form endospores that can survive for millions of years in isolated habitats [1]. Notable examples are bacterial strains isolated and revived from 250-million-year-old salt crystals or from the abdomen of extinct bees preserved in 25–40-million-year-old amber [2,3]. The genomes of such ancient bacteria provide unique opportunities to understand bacterial evolution and metabolic capabilities on a broader scale [4,5,6].

Recently, a bacterial strain (Mal05) was isolated from intact apple seeds stored in a Roman amphora discovered during the excavation of a Roman villa on Elba Island (Italy) that had been destroyed by a fire about 2000 years ago [7]. The seeds were embedded in a hardened soil layer likely generated during the fire, which may explain their exceptional preservation [7]. The bacterial strain Mal05 was isolated into pure culture from the internal embryo tissue of surface-sterilized intact apple seeds and characterized as an aerobic, Gram-positive and rod-shaped *Bacillus subtilis* strain based on the 16S rRNA gene [7]. However, the *B. subtilis* group consists of numerous closely related bacterial species that are indistinguishable based on the 16S rRNA gene alone [8,9]. Hence, genome sequencing was necessary to determine the exact taxonomic affiliation of the ancient *Bacillus* strain.

A characteristic feature of bacteria belonging to the *B. subtilis* group is their large array of secondary metabolites with antimicrobial activity, including three families of cyclic lipopeptides: fengycins, iturins, and surfactins. All of these consist of a polypeptide ring linked to a fatty acid chain. While iturins and surfactins are heptapeptides linked to a β-amino fatty acid chain or β-hydroxy fatty acid chain, respectively, fengycins are decapeptides linked to a β-hydroxy fatty acid chain [10]. These three lipopeptide families also differ in their antimicrobial activities: while fengycins and iturins are well known for their strong antifungal activities, surfactins can be effective against both bacteria and certain fungi [11,12,13,14]. Indeed, in vitro experiments demonstrated an antagonistic interaction between Mal05 and an *Aspergillus niger* (Ascomycota) strain that had been isolated from the same apple seeds [7,15]. Specifically, Mal05 inhibited conidia and spore production of the fungus when grown in co-culture. This antifungal activity was likely due to the production of surfactins, which significantly increased in co-cultures with the fungus compared to mono-cultures of the bacterium [15]. Hence, cyclic lipopeptides are essential for *B. subtilis* to compete against other microbes present in the same environment, making them promising candidates as biocontrol agents against numerous plant pathogens [10,11,13,16]. In addition, surfactins have gained attention for their potential as biotechnological tools, for instance, as natural preservatives for food and beverage products due to their antimicrobial properties (e.g., [17]); as an insecticide against various insect species (e.g., [18]); as an antiviral agent (e.g., [19,20]); and as a biofilm inhibitor (e.g., [21]).

All three cyclic lipopeptide families are synthesized by non-ribosomal peptide synthetases (NRPS). The genes coding for these NRPS are organized in operons of 4–5 genes, namely *fenA-E* for fengycin and *ituA-D* (or *fenF, mycA-C*) for iturins [10]. The five enzymes involved in surfactin biosynthesis are organized in an operon of four genes (*srfAA*, *srfAB*, *srfAC*, and *srfAD*) and a fifth gene, *sfp*, is located about 4 Kbp downstream of the operon [22,23,24]. The importance of these cyclic lipopeptides for the interaction against microbial competitors in both the ancient and modern *Bacillus* strains suggests that this function has been conserved in this genus for a substantial amount of time. However, the selective pressures acting on the underlying genes have not yet been investigated. Specifically, if these compounds are essential for the bacterium’s fitness, the underlying genes would be expected to be under purifying selection to conserve these functions. On the other hand, if competing microorganisms can develop resistance against them, one might expect these genes to be under positive selection to counteract these mechanisms.

Herein, we present the complete genome of the 2000-year-old Mal05 strain obtained using the Oxford Nanopore long-read sequencing technology. Using this genome sequence, the strain was identified as *B. stercoris* based on phylogenomic, Average Nucleotide Identity (ANI), and digital DNA–DNA hybridization (dDDH) analyses. In accordance with previously sequenced *B. stercoris* strains [25], we identified the presence of complete operons for the lipopeptides fengycin and surfactin, while iturins were absent. In addition, we investigated the selective pressures acting on the genes necessary for fengycin and surfactin biosynthesis in this species. The genome of the ancient Mal05 strain greatly enhanced the evolutionary relevance of this analysis by increasing its temporal scale.

## 2. Materials and Methods

### 2.1. Genome Sequencing, Assembly, and Annotation

The *Bacillus* strain Mal05 was kindly provided by Franco Baldi, who had initially isolated it from excavated apple seeds. The strain was grown in Luria Bertani culture medium for 24 h at 30 °C. DNA was extracted using the DNeasy^®^ Blood and Tissue kit (QIAGEN, Hilden, Germany) from 2 mL of bacterial culture. DNA integrity was verified by 0.8% agarose gel electrophoresis at 90 V for 1 h. DNA purity and concentration were measured with a NanoDrop 1000 spectrophotometer (Thermo Fisher Scientific, Waltham, MA, USA) and the Qubit double-stranded DNA high-sensitivity assay kit. 7 μg of the extracted DNA was used for 2 × 150 bp paired-end sequencing on an Illumina NovaSeq (Macrogen Europe, Amsterdam, The Netherlands). 2.5 μg of DNA was used for library preparation using the Oxford Nanopore Ligation Sequencing Kit SQK-LSK 109 (Oxford Nanopore Technologies, Oxford, UK). The library was sequenced on an R9.4 flow cell on a MinION Mk1B sequencer for 48 h using MinKNOW v18.03.1. Basecalling was performed using Guppy v4.4.1 [26] with the high-accuracy algorithm and a minimum quality of Q9. Only reads longer than 500 bp were used for genome assembly.

The long reads were assembled into a single contig using Flye v2.8.1 [27]. This contig was circularized using Circlator v1.5.5 [28] with the options --merge_min_id 85 and --merge_breaklen 1000 as advised for Oxford Nanopore reads. The circular genome was polished with the Illumina reads using POLCA (MaSuRCA v4.0.1) [29,30]. Genome completeness was assessed using BUSCO v4.1.4 [31] with the Bacillales database after the assembly, circularization, and polishing steps. The final genome was automatically annotated using the NCBI Prokaryotic Genome Annotation Pipeline (PGAP) version r2021-01-09.build5126 [32]. Cluster of Orthologous Gene (COG) categories were determined using eggNOG-mapper v2.1.12 [33], and a circular genome plot was produced using MGCplotter (https://github.com/moshi4/MGCplotter).

### 2.2. Phylogenomics

To determine the taxonomy and phylogenetic position of the Mal05 strain, a phylogenomic reconstruction of the *B. subtilis* species complex was performed, including 19 genomes of 15 species and subspecies of the *B. subtilis* group (Appendix A) as well as Mal05. *B. cereus* was used as the outgroup. The genomes of type-strains were chosen whenever high-quality genomes were available for them. When the genome of a given type-strain was of low quality, as in the case of *B. stercoris* (i.e., highly fragmented, high number of pseudogenes), the NCBI reference genome for this species was included as well. A total of 954 single-copy orthologs present in all genomes were identified using OrthoFinder v2.3.3 [34]. The amino acid sequences of each gene were aligned using MUSCLE v3.8.31 [35]. The alignments were concatenated using geneStitcher (https://github.com/ballesterus/Utensils), and the best evolutionary model for each gene was determined using PartitionFinder 2 v2.1.1 [36]. A maximum-likelihood phylogenetic tree was built using RAXML [37], performing 1000 bootstraps with a partitioned maximum-likelihood model applying the evolutionary models assigned to each gene. In addition, the Average Nucleotide Identity (ANI) between the 20 reference genomes and Mal05 was calculated using the ANI calculator [38] implemented in the enveomics toolbox [39]. Digital DNA–DNA hybridization (dDDH) was calculated using TYGS [40] as implemented on the DSMZ website (https://tygs.dsmz.de/; accessed on 20 January 2024).

A second phylogenomic analysis was performed for the species *B. stercoris*, including eight out of the eleven available genomes (as of October 2023), with two *B. subtilis* subsp. *subtilis* genomes as the outgroup. The genomes of the strains D7XPN1, SMPL704, and SMPL712 (Appendix A) were not used due to a high number of pseudogenes, indicating a low genome quality. A maximum-likelihood phylogenetic tree based on 2623 single-copy orthologs was produced as outlined above.

### 2.3. Evolutionary Analyses of the Surfactin Biosynthesis Genes

The gene clusters containing the fengycin and surfactin operons as well as the *sfp* gene were identified in all *B. stercoris* genomes using AntiSMASH [41] and visualized using Clinker [42] implemented on the CAGECAT webserver. The HyPhy suite [43] implemented on the Galaxy platform [44] was used to test whether the 10 genes involved in fengycin and surfactin biosynthesis (i.e., *fenA* to *fenE*, *srfAA* to *srfAD*, and *sfp*) were under specific selective pressure (i.e., purifying or diversifying selection). The FUBAR analysis [45] was used to determine if specific sites (codons) in each gene were under selective pressure, while aBSREL [46] and RELAX [47] were used to infer if specific branches of the *B. stercoris* phylogeny were under selection. All analyses were carried out using default parameters. To test whether the functional domains of each gene were enriched in sites under selection, the proportion of sites under selection in each domain was compared to the proportion of sites under selection across the entire gene using the Z-score test with the function “prop.test” in R v4.1.2 [48].

## 3. Results and Discussion

### 3.1. The Mal05 Strain Belongs to Bacillus stercoris

A hybrid sequencing approach combining Oxford Nanopore long-reads and Illumina short-reads produced a 4.28 Mbp circular genome of the Mal05 strain. It contained 4105 protein-coding genes (CDS), 12 ribosomal rRNA operons, and 94 tRNAs (Figure 1A). The GC content of the genome was 43.3% (Figure 1A). This was very similar to the *B. subtilis* reference genome *B. subtilis* 168, which is 4.22 Mbp long and has 4237 CDS, ten ribosomal operons, 86 tRNAs, and a GC content of 43.5%. However, a phylogenomic analysis of 19 genomes representing 15 species of the *B. subtilis* species complex clearly placed the Mal05 strain within the species *B. stercoris*, a sister species to *B. subtilis* (Figure 2). This was also supported by Average Nucleotide Identity (ANI) and digital DNA–DNA hybridization (dDDH) analyses, since Mal05 had 98.5% ANI and 93.8–94.8% dDDH with two *B. stercoris* strains, compared to only 95% ANI and 80.2–88.6% dDDH with *B. subtilis* and much lower values for all other species (Figure 2). An ANI threshold of 96% was previously used to delineate different species within the *B. subtilis* group [25].

The *B. stercoris* reference genome BS21 was longer than the Mal05 genome (4.78 and 4.28 Mbp, respectively) and contained 4713 CDS compared to 4105 for Mal05. A pangenome analysis based on the distribution of orthogroups between the eight high-quality *B. stercoris* genomes that are currently available revealed that they shared 3320 orthogroups, while 118 orthogroups (i.e., 118 genes) were specific to Mal05 and absent from all other genomes. 44 of these 118 genes were organized into four gene clusters containing 5–17 genes each, albeit without any apparent functional enrichment within these clusters. Hence, despite being highly similar to the other *B. stercoris* genomes, the Mal05 genome also presented a few differences that set it apart. However, it has to be kept in mind that only one of the published genomes was complete (BS21). Therefore, it cannot be excluded that some of these genes are present in other sequenced *B. stercoris* strains, but that they were missed in the current genome assemblies. Moreover, the three genomes (Mal05, BS21, and *B. subtilis* 168) were very similar in terms of functional Clusters of Orthologous Genes (COGs) categories (Figure 1B).

In a second phylogenomic analysis including all eight high-quality *B. stercoris* genomes that are currently available, Mal05 clustered with the PSM7 strain isolated from a landfill site. The cluster formed by Mal05 and PSM7 was in a basal position compared to all the other strains (Figure 3). This basal position made sense, considering the age of the Mal05 strain.

### 3.2. The Fengycin and Surfactin Operons Are Present in All B. stercoris Genomes but Are Not Always Complete

The presence of the two cyclic lipopeptides fengycin and surfactin has been suggested to be characteristic for *B. stercoris* and to distinguish it from other former *B. subtilis* subspecies that possess different sets of lipopeptides (i.e., surfactin alone, surfactin and iturins, surfactin, fengycin, and iturins) [25]. Unfortunately, only two of the eight *B. stercoris* genomes used in this study are currently complete. Therefore, the operons for fengycin (*fenA-E*) and surfactin (*srfAA*, *srfAB*, *srfAC*, *srfAD*, and *sfp*) biosynthesis are not completely present in all genomes. Notably, the genes *srfAA* and *srfAB* of the surfactin operon were fragmented in four of the eight genomes. The fragmented genes were at the extremities of contigs, suggesting that this was rather due to incomplete genome assemblies than being true pseudogenes. Similarly, the genes *fenA*, *fenB*, and in some cases also *fenC* of the fengycin operon were missing in five genomes, with the remaining two or three genes being located at the end of a contig. Consequently, only four genomes (Mal05, BS21, PSM7, and DHFI4) could be included in the subsequent analyses of the surfactin biosynthesis genes, and only three genomes (Mal05, BS21, and DHFI4) for the fengycin biosynthesis genes (Figure 3).

For both the surfactin and fengycin operons, the synteny of these genes as well as the surrounding genomic regions were highly conserved in all genomes in which the relevant genes were complete, except for three genes about 6.7 Kbp upstream of the surfactin operons that are missing in the BS21 strain and two genes upstream of the fengycin operon that only occur in the DHFI4 strain (Figure 4). However, these genes are not involved in the biosynthesis of either lipopeptide. Similar to other described bacilli [22,23,24], the four genes *srfAA*, *srfAB*, *srfAC*, and *srfAD* formed a single operon, and a fifth gene coding for the *sfp* enzyme was located four genes downstream of the operon (Figure 4). Likewise, the five genes *fenA–E* formed a single operon (Figure 5). The *srfA* and *fen* operons encode enzymatic modules that form the non-ribosomal peptide synthetases. Each module consists of numerous domains that incorporate and modify specific amino acids into the peptide ring [10,49].

In addition to the conserved synteny, the nucleotide sequence identity between the three or four strains was also high across the ten genes. As such, the surfactin biosynthesis genes of Mal05 were 97.9–100% identical to the corresponding genes of the three other strains, and the fengycin biosynthesis genes of Mal05 were 96.4–98.8% identical to the corresponding genes of the two other strains.

### 3.3. The Fengycin and Surfactin Biosynthesis Genes Are Not under Strong Selective Pressure

We next investigated whether the five genes involved in surfactin and fengycin biosynthesis, respectively, were under purifying or diversifying selection based on the ratio between non-synonymous and synonymous substitutions (dN/dS). First, we identified the sites (i.e., codons) under selection in each gene (Appendix A). For both lipopeptides, this revealed that (i) the majority of sites in each gene were under neutral selection (92.62–97.32% for surfactin and 94.14–97.62% for fengycin), and (ii) the proportions of the sites under purifying selection were higher than those under diversifying selection (Table 1, Figure 4 and Figure 5). Specifically, 2.68–6.87% of the sites were under purifying selection among the five surfactin genes, whereas the sites under diversifying selection represented 1.11% in *srfAA*, 0.08% in both *srfAB* and *srfAC*, and none in *srfAD* or *sfp* (Table 1, Figure 4 and Figure 5). Similarly, 2.38–5.86% of the sites were under purifying selection among the five fengycin genes, while sites under diversifying selection were only detected in *fenA* and *fenD* (0.12 and 0.03%, respectively) (Table 1). To test whether the sites under selection occurred more frequently within the functional domains compared to other genic regions, the proportion of sites under selection within each functional domain was compared to the proportion of sites under selection across the entire gene. Based on this comparison, the distribution of sites under selection did not differ between the functional domains and other regions in any of the genes (Z-score test, all *p* > 0.05).

In contrast, when investigating the selective pressures acting on the branches of the *B. stercoris* phylogeny, no significant purifying or diversifying selection could be identified for any of the ten genes on any specific branch of the phylogenetic tree. Taking these two results together (i.e., site and branch selection), it appeared that although some sites were under purifying selection, there was no strong selective pressure (either purifying or diversifying) on these genes throughout the evolution of *B. stercoris*. The relatively weak purifying selection could be explained by the fact that neither fengycin nor surfactin are crucial for the survival of *B. stercoris*. In addition, the proportion of sites under diversifying selection was extremely low, suggesting that the two lipopeptides are not involved in an evolutionary arms race with other microorganisms trying to evade them. Indeed, members of the *B. subtilis* group are often able to produce an arsenal of different secondary metabolites (e.g., antimicrobial peptides) to help them compete with other microorganisms in the environment [50,51,52,53]. Nonetheless, surfactin production can still provide a competitive advantage against specific microorganisms, as in the case of Mal05 against a fungus [15]. Both surfactin and fengycin could also be involved in additional functions, such as biofilm formation, plant colonization, and the induction of systemic resistance against pathogens in plants [10,11,16].

To our knowledge, this was the first study investigating the selective pressures acting on two cyclic lipopeptides that play key roles in the *B. subtilis* group. A major limitation of this study was the low number of high-quality *B. stercoris* genomes that are currently available. Therefore, our results need to be interpreted with caution and different patterns may emerge as more high-quality genomes become available in the future. An interesting avenue for future research would be to perform similar analyses in different species of the *B. subtilis* group to investigate whether the neutral selection observed here is a consistent pattern for these genes or whether it depends on the *Bacillus* species. In addition, it would be interesting to perform the same analyses for diverse iturins, as this lipopeptide family is absent from *B. stercoris*.

## 4. Conclusions

Herein, we sequenced the genome of a 2000-year-old *Bacillus* strain isolated from intact apple seeds recovered during archaeological excavations of a Roman villa. Phylogenomic, ANI, and dDDH analyses allowed us to assign this strain to the species *B. stercoris*. Comparative genomic analyses revealed that all sequenced *B. stercoris* strains encoded the biosynthesis genes for the two cyclic lipopeptides fengycin and surfactin, which are known for their antimicrobial activities against fungi and bacteria. Evolutionary analyses demonstrated for the first time that, despite being highly conserved in this species, the biosynthesis genes for both lipopeptides were mainly under neutral selection. An interesting avenue for future research would be to investigate whether this pattern is also observed in other *Bacillus* species possessing different sets of cyclic lipopeptides.

## Figures and Tables

**Figure 1 microorganisms-12-00338-f001:**
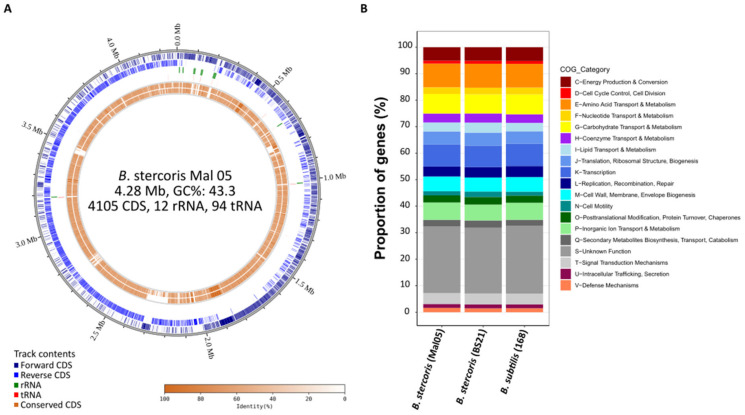
The genome of *Bacillus stercoris* strain Mal05. (**A**) Circular genome plot of *B. stercoris* Mal05. The four outer-most circles represent forward CDS, reverse CDS, ribosomal RNAs, and tRNAs. The inner circles represent the conserved genes in *B. stercoris* strain BS21 and *B. subtilis* strain 168. The shading indicates the degree of sequence similarity compared to *B. stercoris* Mal05. (**B**) Comparison of functional COG categories between *B. stercoris* Mal05, *B. stercoris* strain BS21, and *B. subtilis* strain 168.

**Figure 2 microorganisms-12-00338-f002:**
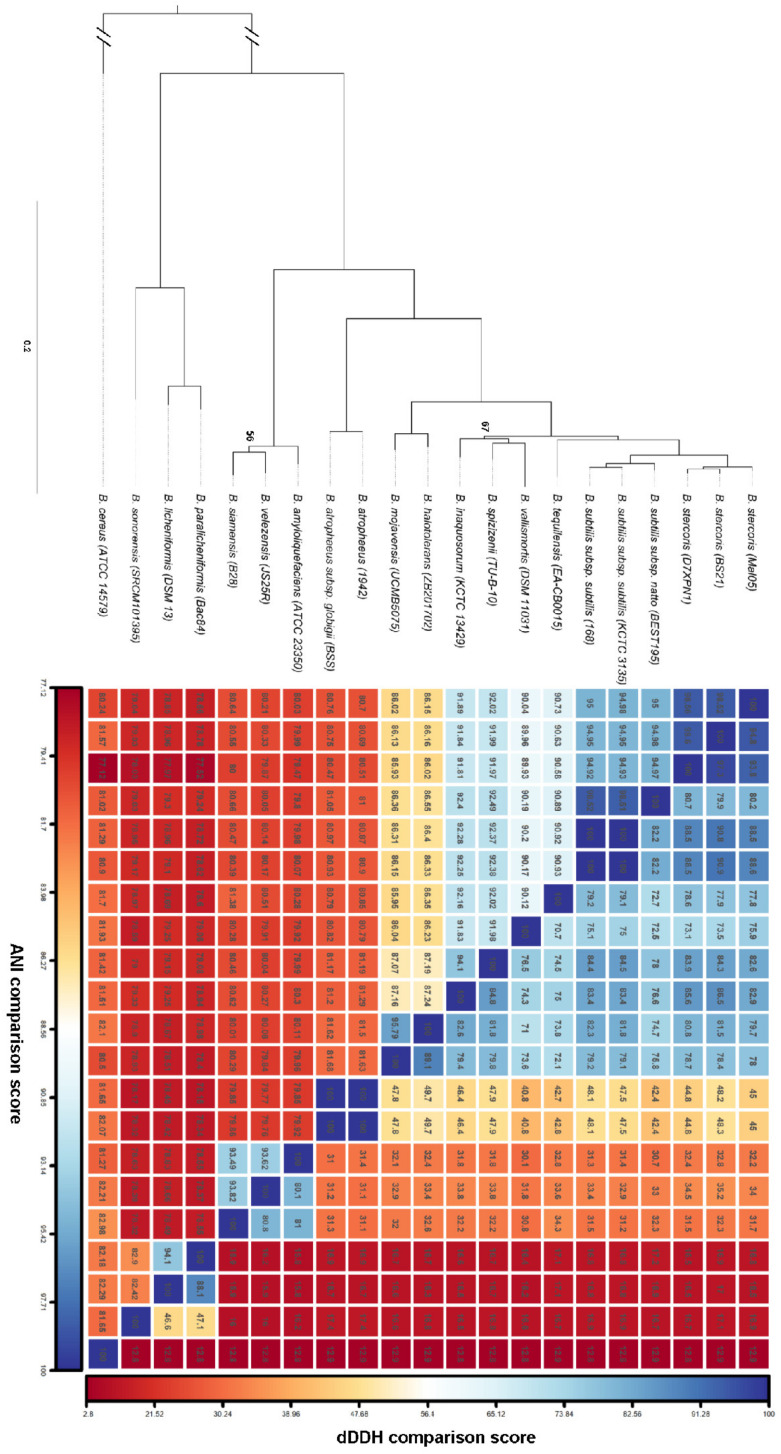
Phylogeny of the *Bacillus subtilis* group. Left panel: maximum-likelihood tree based on the concatenated amino acid sequence alignment of 954 single-copy orthologous genes from 19 reference genomes representing the different species and subspecies of the *B. subtilis* group with *B. cereus* as the outgroup. Only bootstrap values below 100 are indicated. Right panel: a matrix of the Average Nucleotide Identity (ANI) and digital DNA–DNA hybridization (dDDH) values between the different genomes is shown next to the tree. The color scale ranges from dark red (low-similarity values) to dark blue (high-similarity values).

**Figure 3 microorganisms-12-00338-f003:**
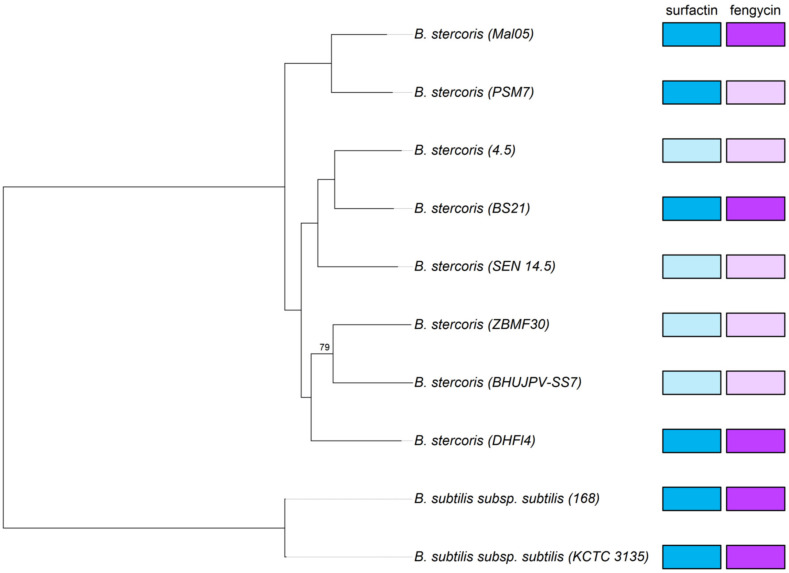
Phylogeny of *B. stercoris*. Maximum-likelihood tree based on 2623 single-copy orthologous genes present in eight high-quality *B. stercoris* genomes with *B. subtilis* subsp. *subtilis* as the outgroup. Only bootstrap values below 100 are indicated. The completeness of the fengycin and surfactin biosynthesis operons in each genome is indicated on the right: dark colors = complete, pale colors = incomplete.

**Figure 4 microorganisms-12-00338-f004:**
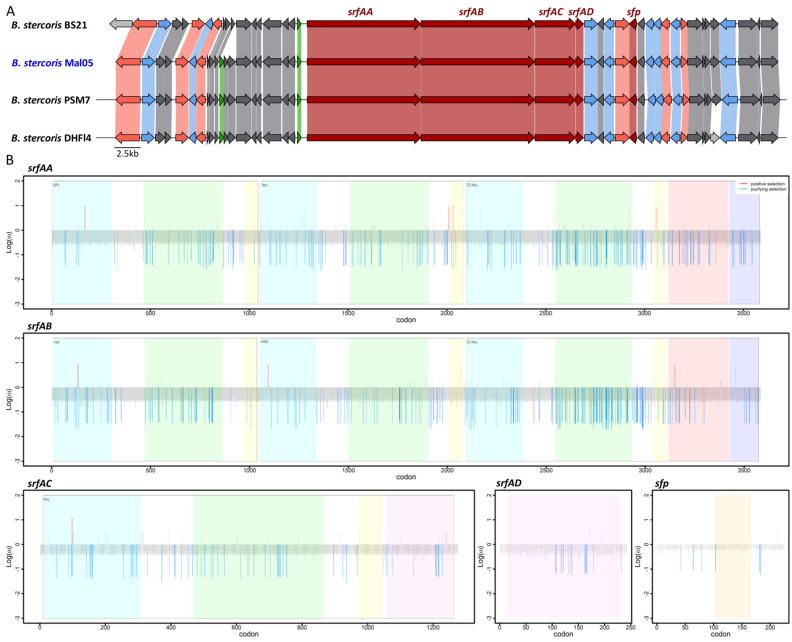
Synteny and evolution of the surfactin biosynthesis genes. (**A**) Synteny of the genomic region surrounding the surfactin biosynthesis genes. The five genes involved in surfactin biosynthesis are colored in dark red. Pink indicates genes involved in other biosynthetic processes, green indicates genes involved in regulation, and blue indicates genes involved in transport. Grey indicates genes involved in other processes. (**B**) Sites under selection for each gene. Blue indicates genes under purifying selection, red indicates sites under diversifying selection. Colored shading delimits the different functional domains in each gene. The x-axis reports the position of each site (i.e., codon) while the y-axis reports the log10 of the dN/dS ratio.

**Figure 5 microorganisms-12-00338-f005:**
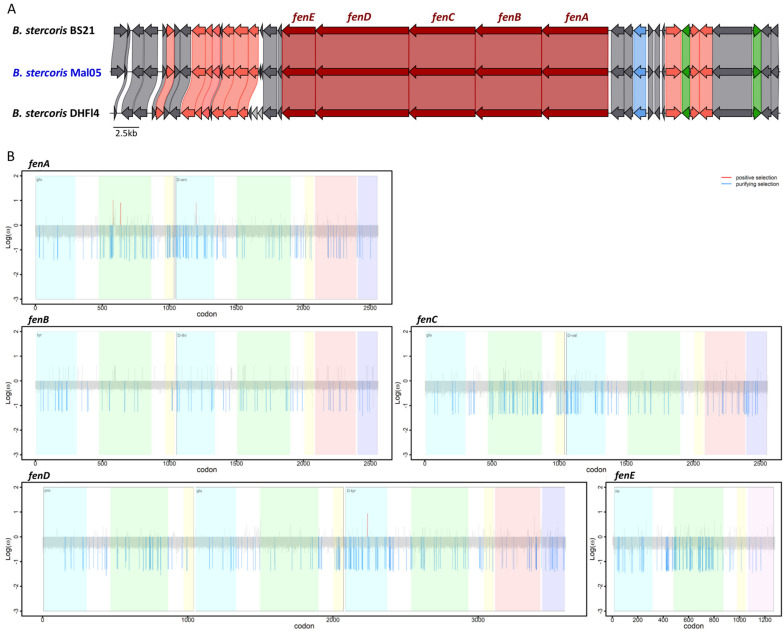
Synteny and evolution of the fengycin biosynthesis genes. (**A**) Synteny of the genomic region surrounding the fengycin biosynthesis genes. The five genes involved in fengycin biosynthesis are colored in dark red. Pink indicates genes involved in other biosynthetic processes, green indicates genes involved in regulation, and blue indicates genes involved in transport. Grey indicates genes involved in other processes. (**B**) Sites under selection for each gene. Blue indicates genes under purifying selection, red indicates sites under diversifying selection. Colored shading delimits the different functional domains in each gene. The x-axis reports the position of each site (i.e., codon) while the y-axis reports the log10 of the dN/dS ratio.

**Table 1 microorganisms-12-00338-t001:** Percentage of codons under purifying, neutral, or diversifying selection in each gene.

Gene	Purifying	Neutral	Diversifying
** *fenA* **	4.64	95.24	0.12
** *fenB* **	2.38	97.62	0
** *fenC* **	4.62	95.38	0
** *fenD* **	4.52	95.45	0.03
** *fenE* **	5.86	94.14	0
** *srfAA* **	6.27	92.62	1.11
** *srfAB* **	6.87	93.05	0.08
** *srfAC* **	4.56	95.36	0.08
** *srfAD* **	4.96	95.04	0
** *sfp* **	2.68	97.32	0

## Data Availability

The Mal05 genome was deposited in the NCBI database under Bioproject number PRJNA952856. The genome described in this publication corresponds to the first version of the submitted genome.

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
