# Peer review of "A 2000-Year-Old Bacillus stercoris Strain Sheds Light on the Evolution of Cyclic Antimicrobial Lipopeptide Synthesis"

_microorganisms, 2024, doi:10.3390/microorganisms12020338_

Round 1

Reviewer 1 Report

Comments and Suggestions for Authors

Author Response

Reviewer 1

Generally speaking, lipopeptides include fengycin, iturin, and surfactin. Among them, fengycin and iturin are considered the primary antibacterial compounds. Surfactin’s ability to inhibit fungal activity is not very strong. According to the genome of Mal05, 11 types of secondary metabolite gene clusters can be predicted using antiSMASH, including fengycin. Therefore, why didn’t this study choose fengycin cluster for stress selection analysis? Perhaps fengycin is essential in the evolutionary process.

- Our initial choice to focus only on surfactin was motivated by experimental data demonstrating an increase in surfactin (but not fengycin) in co-cultures with a fungal strain (ll. 53-58). Nonetheless, we completely agree with the reviewer that it is more interesting to include fengycin as well to obtain a more complete picture of the evolutionary processes acting on these lipopeptides. Hence, we performed the same analyses also on the five genes involved in fengycin biosynthesis and added new information regarding the structure and function of fengycin and iturin in the introduction of the revised version (ll. 45-53).

Line 57 How many amino acids does surfactin contain?

- Seven, this information was added in the revised version (l. 48).

Lines 59-60 Gene names should be in italics, please carefully check the manuscript.

- Done

Line 81 “24h” should be “24 h”.

- Done

Line 82 “2 ml” should be “2 mL”.

- Done

Line105 Are the strains used in figure 2 type strains(https://lpsn.dsmz.de/)?

If not, please re-construct the figure 2.

- We have repeated this analysis using type strains whenever high-quality genomes were available for them. Unfortunately, this was not always the case. When the genome of a given type strain was of low quality (i.e. highly fragmented, high number of pseudogenes), we decided to use the NCBI reference genome for this species instead. This choice was motivated by the fact that it is better to use genomes with the highest quality for such analyses to avoid artefacts in the phylogeny. Information on the genomes used and their status (type strain or not) can be found in Supplementary Table S1.

Figure 2 is not clear and needs to be improved in resolution.

- We produced a new figure 2 with better resolution.

Figure 2 lacks dDDH (digital DNA-DNA hybridization) data, which is an important piece of evidence for bacterial classification. Generally, it should appear alongside ANI (Average Nucleotide Identity). Please supplement the dDDH data, which should also be presented in the form of a heatmap.

- Done, dDDH was added to the figure as suggested.

List the dN/dS, dN and/or dS values separately.

- The dN and dS values for each codon of the 10 investigated genes are now provided in the new Supplementary table S2.

References 4 and 44 are not searchable in Google Scholar.

- We added the DOI for the first reference in the revised version to make it more easily searchable. Unfortunately, we could not find a DOI for the second reference. However, we verified that it can be found on Google Scholar.

Reviewer 2 Report

Comments and Suggestions for Authors

The communication article titled "A 2000-year-old Bacillus stercoris strain sheds light on the evolution of surfactin synthesis" provides an intriguing glimpse into the microbial world by examining a bacterial strain isolated from intact apple seeds preserved for two millennia. The study explores the evolutionary aspects of surfactin synthesis, a topic of interest due to its potential applications in various fields.

Here is a list of specific items to improve in the article:

-The methods section is thorough, detailing the genome sequencing, assembly, and annotation processes. The inclusion of phylogenomic analyses, Average Nucleotide Identity (ANI) calculations, and evolutionary analyses of surfactin biosynthesis genes adds depth to the study. However, the lack of detailed information on the specific settings and parameters used in the evolutionary analyses may limit the reproducibility of the results.

-Explicitly address potential limitations in the study, such as the small sample size of high-quality B. stercoris genomes. Acknowledge the need for caution in generalizing the findings.

-Discuss the broader implications of the findings. How do these results contribute to our understanding of bacterial evolution, especially in the context of antimicrobial production?

-Emphasize the caution needed in generalizing findings based on the limited number of B. stercoris genomes analyzed. Mention that different patterns may emerge as more genomes become available.

-Ensure that the conclusion effectively summarizes the key findings and their significance. Consider rephrasing to make the conclusion more engaging for the reader.

-Include a brief discussion on potential avenues for future research. What questions remain unanswered, and how could further studies build on the current findings?

Author Response

Reviewer 2

The communication article titled "A 2000-year-old Bacillus stercoris strain sheds light on the evolution of surfactin synthesis" provides an intriguing glimpse into the microbial world by examining a bacterial strain isolated from intact apple seeds preserved for two millennia. The study explores the evolutionary aspects of surfactin synthesis, a topic of interest due to its potential applications in various fields.

We thank the reviewer for the positive feedback.

Here is a list of specific items to improve in the article:

-The methods section is thorough, detailing the genome sequencing, assembly, and annotation processes. The inclusion of phylogenomic analyses, Average Nucleotide Identity (ANI) calculations, and evolutionary analyses of surfactin biosynthesis genes adds depth to the study. However, the lack of detailed information on the specific settings and parameters used in the evolutionary analyses may limit the reproducibility of the results.

All evolutionary analyses were performed using default settings, unless otherwise stated. This is mentioned in the Materials and Methods (ll. 143-144).

-Explicitly address potential limitations in the study, such as the small sample size of high-quality B. stercoris genomes. Acknowledge the need for caution in generalizing the findings.

Done, this information can be found in ll. 283-286.

-Discuss the broader implications of the findings. How do these results contribute to our understanding of bacterial evolution, especially in the context of antimicrobial production?

A paragraph to this effect is present in the discussion (ll. 270-277).

-Emphasize the caution needed in generalizing findings based on the limited number of B. stercoris genomes analyzed. Mention that different patterns may emerge as more genomes become available.

Done, this information can be found in ll. 283-286.

-Ensure that the conclusion effectively summarizes the key findings and their significance. Consider rephrasing to make the conclusion more engaging for the reader.

The conclusion was rephrased in the revised version (ll. 293-302).

-Include a brief discussion on potential avenues for future research. What questions remain unanswered, and how could further studies build on the current findings?

This information was added to the discussion (ll. 286-291).

Reviewer 3 Report

Comments and Suggestions for Authors

The presented manuscript discusses very interesting material obtained from sequencing the complete genome of Bacillus stercoris strain Mal05, preserved for 2000 years in the form of spores. I agree with the authors' reclassification of the strain from Bacillus subtilis to Bacillus stercoris based on the whole genome sequence.

However, I have several major remarks about the presented text:

1. Classification as a species can be made only after comparing the strain under study with type strains of closely related species (for this work, Bacillus subtilis and Bacillus stercoris). For the species Bacillus subtilis, the type strain is Bacillus subtilis KCTC3135. For the species Bacillus stercoris, the type strain is Bacillus stercoris D7XPN1. But the authors (Figures 1 and 2), without any justification, compare the genome of strain Mal05 with strains of Bacillus subtilis 168, which is not a type strain (https://bacdive.dsmz.de/strain/1003). And instead of the Bacillus stercoris D7XPN1 genome, the Bacillus stercoris BS21 genome is used (Figure 2), explaining the large number of pseudogenes in the Bacillus stercoris D7XPN1 genome. I completely disagree with this, since using a genome with pseudogenes will only lead to a decrease in the number of orthologous genes that the authors use for phylogenetic analysis. But the remaining single-copy orthologous genes will be quite sufficient for correct taxonomic analysis. While determining the species of a strain in comparison with non-type strains is nonsense. Use type strains of Bacillus spp. for phylogenomic analysis (Figure 2).

2. The article describing the species Bacillus stercoris (https://doi.org/10.1007/s10482-019-01354-9) states that the taxonomic feature of these bacteria is the genes to produce fengycin. Fengycin is also a lipopeptide similar to surfactin. However, the authors do not mention the genes of fengycin biosynthesis. Why were the genes for the biosynthesis of surfactin, but not fengycin, chosen for analysis?

3. The authors limit themselves to comparing the number of CDSs in genomes. However, no information is provided about the presence in the genome of Bacillus stercoris Mal05 of any specific genes or gene clusters that are absent in other closely related strains. Conversely, Figures 3 and 4A show the similarity of the five genes involved in surfactin biosynthesis, but there is no data from multiple alignments of these genes, information about the identity of the nucleotide sequences or the amino acid sequences of the protein products. As a result, I don't understand why the Bacillus stercoris Mal05 genome is interesting. The authors show that surfactin biosynthesis genes are not affected by selection, but do not provide genes that are affected by selection for comparison. One can conclude that selection does not act on Bacillus at all, but the authors do not indicate this either.

Minor remarks:

1. Line 60: “about 5 Kbp downstream” – why 5 Kbp? The mentioned references describe 4 Kbp. In the genome of strain Mal05, this distance is 4343 bp.

2. Lines 158-160: “The cluster formed by Mal05 and PSM7 is in a basal position compared to all other strains (Figure 3). This basal position makes sense, considering the age of the Mal05 strain.” - Does this mean that the PSM7 strain is also very old?

3. Lines 187-191: How statistically significant are the differences between genes encoding multiple functional domains and genes with a single functional domain? Moreover, srfAC has 4.56%, and srfAD has 4.96%.

Author Response

Reviewer 3

The presented manuscript discusses very interesting material obtained from sequencing the complete genome of Bacillus stercoris strain Mal05, preserved for 2000 years in the form of spores. I agree with the authors' reclassification of the strain from Bacillus subtilis to Bacillus stercoris based on the whole genome sequence.

We thank the reviewer for the positive feedback.

However, I have several major remarks about the presented text:

  1. Classification as a species can be made only after comparing the strain under study with type strains of closely related species (for this work, Bacillus subtilis and Bacillus stercoris). For the species Bacillus subtilis, the type strain is Bacillus subtilis KCTC3135. For the species Bacillus stercoris, the type strain is Bacillus stercoris D7XPN1. But the authors (Figures 1 and 2), without any justification, compare the genome of strain Mal05 with strains of Bacillus subtilis 168, which is not a type strain (https://bacdive.dsmz.de/strain/1003). And instead of the Bacillus stercoris D7XPN1 genome, the Bacillus stercoris BS21 genome is used (Figure 2), explaining the large number of pseudogenes in the Bacillus stercoris D7XPN1 genome. I completely disagree with this, since using a genome with pseudogenes will only lead to a decrease in the number of orthologous genes that the authors use for phylogenetic analysis. But the remaining single-copy orthologous genes will be quite sufficient for correct taxonomic analysis. While determining the species of a strain in comparison with non-type strains is nonsense. Use type strains of Bacillus spp. for phylogenomic analysis (Figure 2).

Figure 1 reports a functional comparison between closely-related genomes, its purpose is not to assign strain Mal05 to any of the two species. For a functional comparison to be meaningful, complete and high-quality genomes are necessary. Therefore, we used the reference genomes for both B. subtilis and B. stercoris here, although they are not from the type strains. Regarding B. subtilis, the type strain KCTC3135 is actually included in Figure 2 in addition to the reference genome 168 and both genomes share 100% ANI (Fig. 2), they are therefore highly equivalent.

The genome of the B. stercoris type strain D7XPN1 is more problematic. First of all, the genome is incomplete, therefore we cannot make any meaningful comparisons in terms of functional content or even genome size compared to our genome and need to use the reference genome BS21 for the first paragraph of the result section, as this genome is complete. Moreover, the fact that approximately half of the genes in the D7XPN1 genome are pseudogenes means that there was a problem with the sequencing or the assembly, introducing a high number of errors in the final genome. Using such a poor-quality genome in phylogenetic analyses will produce erroneous results because even the remaining genes likely contain sequence errors, which may lead to poorly supported nodes, long branching and other artifactual results that are mainly due to the quality of the genome rather than their real evolution. Therefore, we still decided to use the strain BS21 instead of D7XPN1 for these analyses.

Nonetheless, we have redone the phylogenomic analysis in Figure 2 to include genomes of type strains whenever those strains possessed high-quality genomes. When their genomes were of low quality, we decided to use the NCBI reference genome instead. This choice was motivated by the fact that it is better to use genomes with the highest quality for such analyses to avoid artefacts in the phylogeny. Information on the genomes used and their status (type strain or not) can be found in Supplementary Table S1.

  1. The article describing the species Bacillus stercoris (https://doi.org/10.1007/s10482-019-01354-9) states that the taxonomic feature of these bacteria is the genes to produce fengycin. Fengycin is also a lipopeptide similar to surfactin. However, the authors do not mention the genes of fengycin biosynthesis. Why were the genes for the biosynthesis of surfactin, but not fengycin, chosen for analysis?

We thank the reviewer for pointing this out. Our initial choice to focus only on surfactin was motivated by experimental data demonstrating an increase in surfactin (but not fengycin) in co-cultures with a fungal strain (ll. 53-58). Nonetheless, we completely agree with the reviewer that it is more interesting to include fengycin as well to obtain a more complete pictures of the evolutionary processes. Hence, we performed the same analyses also on the five genes involved in fengycin biosynthesis and added new information regarding the structure and function of fengycin and iturin in the introduction of the revised version (ll. 45-53).

  1. The authors limit themselves to comparing the number of CDSs in genomes. However, no information is provided about the presence in the genome of Bacillus stercoris Mal05 of any specific genes or gene clusters that are absent in other closely related strains. Conversely, Figures 3 and 4A show the similarity of the five genes involved in surfactin biosynthesis, but there is no data from multiple alignments of these genes, information about the identity of the nucleotide sequences or the amino acid sequences of the protein products. As a result, I don't understand why the Bacillus stercoris Mal05 genome is interesting. The authors show that surfactin biosynthesis genes are not affected by selection, but do not provide genes that are affected by selection for comparison. One can conclude that selection does not act on Bacillus at all, but the authors do not indicate this either.

We included the results of a pangenome analysis including the number of specific orthogroups that are absent from all other B. stercoris genomes in the revised version (ll. 164-169). Information regarding nucleotide sequence identities for the genes involved in surfactin and fengycin biosynthesis was also included (ll. 221-225).

Regarding the selective pressures, the inclusion of the fengycin biosynthesis operon in the revised version allowed us to show that both cyclic lipopeptides are mainly under neutral selection. This result is interesting as it indicates that these genes may not be as essential for the bacteria as one might expect. In comparison, genes that are involved in essential pathways (e.g. gyrases or topoisomerases) are generally under purifying selection. However, the fact that two operons are not under any significant selective pressure does not allow any extrapolation regarding the rest of the genome. It would be very strange if no genes were under significant selective pressure in Bacillus stercoris. However, applying these analyses to all genes in the genome would be beyond the scope of this paper.

Minor remarks:

  1. Line 60: “about 5 Kbp downstream” – why 5 Kbp? The mentioned references describe 4 Kbp. In the genome of strain Mal05, this distance is 4343 bp.

This mistake was corrected in the revised version (l. 69).

  1. Lines 158-160: “The cluster formed by Mal05 and PSM7 is in a basal position compared to all other strains (Figure 3). This basal position makes sense, considering the age of the Mal05 strain.” - Does this mean that the PSM7 strain is also very old?

Based on the information available in the NCBI genomes database, the strain PSM7 was collected in 2018 from a landfill site. Hence, it does not seem that this strain is very old. However, without additional information, we prefer not to speculate as to why this particular strain is in a basal position. It is likely simply the most-closely related strain to Mal05 among those sequenced to date. Once more high-quality genomes are available, the phylogeny of B. stercoris will become much clearer.

  1. Lines 187-191: How statistically significant are the differences between genes encoding multiple functional domains and genes with a single functional domain? Moreover, srfAC has 4.56%, and srfAD has 4.96%.

This statement was removed in the revised version.

Round 2

Reviewer 2 Report

Comments and Suggestions for Authors

The authors adhered to the previous suggestions and comments provided by the reviewer. The scientific names of bacteria must be written in italics. Please ensure these corrections are properly implemented throughout the manuscript.

Author Response

* The authors adhered to the previous suggestions and comments provided by the reviewer. The scientific names of bacteria must be written in italics. Please ensure these corrections are properly implemented throughout the manuscript.

- We double-checked the entire manuscript and corrected the missing italics.

Reviewer 3 Report

Comments and Suggestions for Authors

The manuscript microorganisms-2743953 has been significantly improved by the authors. From the submitted responses and corrections to the manuscript, I was not convinced by the authors' response in explaining their refusal to use the genome of Bacillus stercoris type strain D7XPN1 in the phylogenetic analysis. Without comparing the genomes of strains Mal05 and D7XPN1, one can only assume, but not confirm, that the strain belongs to the species Bacillus stercoris. Or, the authors should provide additional reasons why they believe that strain Mal05 is Bacillus stercoris, but not Bacillus subtilis. Comparison of the genomes of strains Mal05 and BS21 cannot be used for the taxonomy. Based on what evidence (other than the name in GenBank) do the authors believe that strain BS21 belongs to the species Bacillus stercoris? I assume that for this purpose, the genome of Bacillus stercoris type strain D7XPN1 was used, which the authors do not want to use in this work. In addition, this part of the manuscript should explain (using a reference) why “95% ANI and 80.2-88.6% dDDH with B. subtilis” indicates that the strain is not a B. subtilis species. I do not dispute that strain Mal05 is a member of the species Bacillus stercoris. But the results presented are not convincing. All arguments and doubts of the authors regarding the phylogenetic determination of the Mal05 strain should be in the text of the manuscript (Lines 156–161).

There are also technical mistakes that need to be corrected:

Line 116: more correct would be “18 genomes of 15 species and subspecies of the B. subtilis group”.

Line 154: "...GC content of the genome is 43.4% (Figure 1A)." But Figure 1A shows “GC%: 43.3”.

Author Response

* The manuscript microorganisms-2743953 has been significantly improved by the authors. From the submitted responses and corrections to the manuscript, I was not convinced by the authors' response in explaining their refusal to use the genome of Bacillus stercoris type strain D7XPN1 in the phylogenetic analysis. Without comparing the genomes of strains Mal05 and D7XPN1, one can only assume, but not confirm, that the strain belongs to the species Bacillus stercoris. Or, the authors should provide additional reasons why they believe that strain Mal05 is Bacillus stercoris, but not Bacillus subtilis. Comparison of the genomes of strains Mal05 and BS21 cannot be used for the taxonomy. Based on what evidence (other than the name in GenBank) do the authors believe that strain BS21 belongs to the species Bacillus stercoris? I assume that for this purpose, the genome of Bacillus stercoris type strain D7XPN1 was used, which the authors do not want to use in this work. In addition, this part of the manuscript should explain (using a reference) why “95% ANI and 80.2-88.6% dDDH with B. subtilis” indicates that the strain is not a B. subtilis species. I do not dispute that strain Mal05 is a member of the species Bacillus stercoris. But the results presented are not convincing. All arguments and doubts of the authors regarding the phylogenetic determination of the Mal05 strain should be in the text of the manuscript (Lines 156–161).

- To be honest, we do not understand why the reviewer insists on including a low-quality genome in the analysis (e.g., CheckM completeness 78.7%), unless there are legitimate reasons to believe that the BS21 strain was not assigned to the correct species. Nonetheless, we updated the phylogeny in Figure 2 to include D7XPN1. ANI and dDDH analysis are recognized methods to assign genomes to existing species and in this case, ANI and dDDH values are much higher between Mal05 and B. stercoris BS21 than Mal05 and B. subtilis. Therefore, it would not make sense to assume that the Mal05 strain belongs to B. subtilis with whom it shares lower ANI and dDDH values than with B. stercoris.

We added a sentence that a threshold of 96% ANI was previously used to delineate species within the genus Bacillus (e.g. in Dunlap et al 2020 to differentiate four former subspecies of B. subtilis) (ll. 166-167). We explain our reasoning for the exclusion of low-quality genomes in the Materials and Methods (ll. 118-120 and ll. 134-136).

* There are also technical mistakes that need to be corrected:

Line 116: more correct would be “18 genomes of 15 species and subspecies of the B. subtilis group”.

- The sentence was modified as suggested.

Line 154: "...GC content of the genome is 43.4% (Figure 1A)." But Figure 1A shows “GC%: 43.3”.

- The GC content was corrected in the text.